# Anti-Inflammatory Effects of GLP-1R Activation in the Retina

**DOI:** 10.3390/ijms232012428

**Published:** 2022-10-17

**Authors:** Alessandra Puddu, Davide Maggi

**Affiliations:** Department of Internal Medicine and Medical Specialties, University of Genoa, 16132 Genoa, Italy

**Keywords:** GLP-1, GLP-1R, GLP-1RAs, diabetic retinopathy, glaucoma, age-related macular degeneration, retinal ganglion cells, retinal pigment epithelial cells, endothelial cells

## Abstract

Glucagon-like peptide-1 (GLP-1) is an incretin hormone, mainly produced by enteroendocrine L cells, which participates in the regulation of glucose homeostasis, and in reduction in body weight by promoting satiety. Actions of GLP-1 are mediated by activation of its receptor GLP-1R, which is widely expressed in several tissues including the retina. The effects of GLP-1R activation are useful in the management of type 2 diabetes mellitus (T2DM). In addition, the activation of GLP-1R has anti-inflammatory effects in several organs, suggesting that it may be also useful in the treatment of inflammatory diseases. Inflammation is a common element in the pathogenesis of several ocular diseases, and the protective effects of treatment with GLP-1 emerged also in retinal diseases. In this review we highlight the anti-inflammatory effects of GLP-1R activation in the retina. Firstly, we summarized the pathogenic role of inflammation in ocular diseases. Then, we described the pleiotropic effects of GLP-1R activation on the cellular components of the retina which are mainly involved in the pathogenesis of inflammatory retinal diseases: the retinal ganglion cells, retinal pigment epithelial cells and endothelial cells.

## 1. Introduction

The glucagon-like peptide-1 (GLP-1) is an incretin hormone mainly secreted by the enteroendocrine L cells in the distal intestine [1]. GLP-1 participates in the regulation of glucose homeostasis, and in reduction in body weight by promoting satiety. Therefore, its effects are useful in the management of type 2 diabetes mellitus (T2DM) [2]. Actions of GLP-1 are mediated by interaction with its receptor GLP-1R, a G-protein-coupled receptor originally identified in the endocrine pancreas and later in other tissues [3,4]. GLP-1R activation induces several cellular pathways that overall lead to improved cellular function and survival [3,5]. Briefly, the biological effects of GLP-1R activation are mediated by caveolin-1, the principal component of caveolae, which causes small invagination of the plasma membrane involved in regulating receptor trafficking and in assembling signaling complexes [6,7]. Indeed, activation of GLP-1R is affected by spatiotemporal regulation, which is closely related to receptor trafficking [8]. The consequent intracellular signaling pathways are then determined by the assembling of different downstream effectors, such as cAMP and β-arrestins [9].

The half-life of GLP-1 is very short due to rapid inactivation by the dipeptidyl peptidase 4 (DPP-IV). Therefore, several strategies have been developed to prolong GLP-1R activation, including use of exendin-based therapies, analogues of human GLP-1, and DPP-4-resistant analogues [10] (Table 1). Actually, therapeutic agents such as GLP-1R agonists (GLP-1RAs) and DPP-IV inhibitors are largely employed in the treatment of type 2 diabetes and obesity [3,4,5,11,12,13].

The wide expression of GLP-1R implies that GLP-1 may have a large spectrum of actions. Consequently, benefits of GLP-1R activation are not only limited to improved control of glucose homeostasis [14]. For instance, the activation of GLP-1R has anti-inflammatory effects in several organs, suggesting that it may be also useful in the treatment of inflammatory diseases [15,16].

The first evidence of the anti-inflammatory proprieties of GLP-1RAs came from studies on pancreatic β cells; they have also been observed in the heart, vascular system, liver, kidney, brain, and eye [17,18,19,20]. It has been reported that GLP-1RAs may regulate several proinflammatory pathways, including oxidative stress, cytokine production and recruitment of immune cells [18]. GLP-1RAs may ameliorate inflammatory states through a direct mechanism, by regulating immune cells expressing GLP-1 receptors, or by an indirect mechanism through improvement of glycemic control and weight loss [14]. In particular, activation of GLP-1R leads to reduction in macrophage infiltration, and in expression and secretion of proinflammatory citokynes both in vivo and in vitro [21].

The same anti-inflammatory effects have been reported after inhibition of DPP-IV [22]. It has been shown that also DPP-IV inhibitors decrease inflammatory biomarkers, reducing risk of atherosclerosis and cardiovascular events [23,24]. Moreover, treatment with DPP-IV inhibitors prevents breakdown of the blood–retinal barrier (BRB) and reduces neuronal apoptosis [25].

Interestingly, both GLP-1RAs and DPP-IV inhibitors significantly reduce inflammation independently of weight loss or glycemic control [25,26].

This evidence supports that GLP-1 signaling plays an important role not only in the management of diabetes, but also in the treatment of other diseases. This includes neuronal and cardiovascular diseases, such as Parkinson’s Disease and atherosclerosis [24]. In the last decade, the protective effects of treatment with GLP-1 have emerged also in ocular diseases, especially in diabetic retinopathy and glaucoma. However, this topic has not yet been sufficiently covered.

In this review we highlight the anti-inflammatory effects of GLP-1R activation in retina. Firstly, we summarized the pathogenic role of inflammation in ocular diseases. In addition, we described the pleiotropic effects of GLP-1, GLP-1RAs and DPP-IV inhibitors on the main cellular components of the retina.

## 2. Inflammation in Ocular Disease

Inflammation is a common element in the pathogenesis of several retinal diseases, such as diabetic retinopathy (DR), age-related macular degeneration (AMD), and glaucoma [27,28,29,30]. Moreover, inflammation is often associated with blood–retinal barrier (BRB) breakdown [31] leading to vascular hyperpermeability and tissue edema, which may have vision loss as a final consequence [32]. Indeed, alteration of the BRB is due to the rise of inflammatory cytokines, that, in turn, worsens inflammation by weakening tight junctions, increasing leukocyte adhesion and retinal cell death [28,33,34].

Probably, the high susceptibility of the retina to inflammation is due to the large availability of glucose and its oxidation. This consequence is more evident in diabetic subjects, where levels of oxidative stress are increased, but the activity of antioxidant enzymes is reduced [35,36]. Indeed, the increased flux of glucose induces a chronic proinflammatory environment that progressively leads to onset of DR, one of the most common microvascular complications in diabetic patients. Briefly, in the early stage of DR, underlying chronic inflammation causes progressive alteration of the retinal capillary system, leading to the appearance of microaneurysms and exudates. Then, in the proliferative stage (PDR), the release of inflammatory proangiogenic cytokines, such as vascular endothelial growth factor-A (VEGF-A), promotes aberrant neovascularization in response to hypoxia [30,37,38].

Chronic inflammation induced by age-related oxidative stress plays a crucial role in AMD [30], a progressive degeneration of the macula that leads to central vision impairment [39]. Moreover, activation of macrophages and microglia, with consequent release of proinflammatory cytokines, contributes to worsening retinal degeneration in AMD [40]. Like DR, AMD is classified into two clinical forms: the early stage, characterized by the accumulation of drusen in the subretinal space, which may lead to progressive degeneration of photoreceptors, and dysfunction of the retinal pigment epithelium (RPE); and the late stage, which can be neovascular or non-neovascular [41,42]. In the first case, the increased amounts of VEGF-A, mainly produced by RPE cells, lead to breakdown of the external BRB and the formation of new blood vessels in the retinal tissue.

Neuroinflammatory pathways have an important role in the pathogenesis of glaucoma, the leading global cause of irreversible blindness [27]. Glaucoma is an optic neuropathy characterized by degeneration of retinal ganglion cells (RGCs) and progressive optic nerve damage [27,43]. Indeed, some evidence demonstrates that the activation of macrophages and microglial may directly contribute to RGC death and optic nerve degeneration in glaucoma [40,43,44].

Inflammation is also involved in pathogenesis of retinitis pigmentosa (RP), a genetic retinal disease characterized by the progressive loss of photoreceptors, which results in vision loss [45]; and in pathogenesis of uveitis, an autoimmune disease characterized by inflammation of the uvea, a part of the eye that consists of the iris, ciliary body and choroid [46].

## 3. Anti-Inflammatory Effects of GLP-1 in the Retina

The majority of evidence about anti-inflammatory effects of GLP-1R activation is derived from studies on DR. Several experimental studies demonstrated that GLP-1 and its agonists may prevent the onset of DR or reduce its progression through antiapoptotic and anti-inflammatory mechanisms [47,48,49,50,51]. Exendin-4 prevents BRB breakdown, decreases levels of ICAM-1 in the retina of Goto-Kakizaki(GK) rats and decreases the expressions of PLGF and VEGF and the activation of AKT in vitro [47]. Moreover, treatment with exendin-4 decreases retinal cell death and ROS production in Streptozotocin rats and protects retinal precursor cells from oxidative stress [51]. Both systemic and topical administration of native GLP-1 or GLP-1RAs prevents glial activation and apoptosis in retina of db/db mice [48]. Similar results have been achieved with treatment with DPP-IV inhibitors [31,52]. This goal has been reached with systemic, as well with intravitreous or topical administration of the drugs. In the latter case, this was also performed without affecting blood glucose levels [48], suggesting that GLP-1R activation may have a protective role beyond incretin effects. This is also supported by evidence that neuroinflammation that characterizes glaucomatosis in response to ocular hypertension may be prevented by the GLP-1 agonist NLY01 [53]. Besides the promising preclinical results, clinical trials addressed to evaluate the safety of GLP-1RAs either did not report or found neutral effects of this drugs on DR, except for the controversial results obtained after administration of semaglutide, which seems to rise the rate of DR [54,55,56,57,58].

Beneficial effects of activation of GLP-1R in diabetic models involve both functional and morphological aspects, and result in prevention of BRB breakdown [47], preservation of retinal thickness [59], inhibition of macrophages infiltration and activation [60], prevention of retinal neurodegeneration [53,58,61], of loss of pericytes [62], and of the loss of b-wave amplitude [62,63,64].

Intracellular mechanisms through which GLP-1 may exert anti-inflammatory effects include: decreased expression of glial fibrillary acidic protein [59,65], increased expression of sirtuins [51], prevention of upregulation of apoptotic markers and intracellular adhesion molecules [47,48,58,65,66,67], activation of survival pathways [48,61,68,69], prevention of downregulation of tight junctions [47], decreased release of proinflammatory cytokines [24,47,48,52], maintenance of the antioxidant defense system [66,70].

Potentially, GLP-1R may be activated in all parts of the retina. Indeed, GLP-1R has been found widely expressed in cellular components of the retina, with no significant differences between diabetic or nondiabetic subjects [48,58,63,69,71]. However, GLP-1R expression has not been detected in the eyes of subjects affected by advanced stages of PDR [72]. Interestingly, in the human retina GLP-1 is mainly localized in the ganglion cell layer (GCL) [48], whereas levels of GLP-1 expression have been found reduced in the retinas of diabetic patients in comparison with healthy subjects [48].

Here, we reported the anti-inflammatory effects of GLP-1R activation in the cellular components of the retina which are mainly involved in the pathogenesis of inflammatory retinal diseases.

## 4. Retinal Ganglion Cells and Muller Cells

Retinal ganglion cells (RGCs) are a group of neurons with different size, function and morphology located near the inner surface of the retina. Their long axons extend into the brain, thus connecting the visual information originating from photoreceptors to the central nervous system.

RGCs are closely associated with astrocytes, which are responsible of the inflammation of the ganglion cell layer of the retina [73]. Indeed, the release of the proinflammatory cytokines interleukin-1α (IL-1α), tumor necrosis factor α (TNF-α), and the complement component 1q (C1q) induces the A1 proinflammatory form of astrocytes [74]. All these cytokines are involved in the progression of glaucoma [75,76,77].

Due to their importance in visual production, damage of the RGCs may lead to visual impairment and even to permanent loss of vision [78]. In diabetes, neurodegeneration occurs earlier than the microvascular injury of the retina, and it is well recognized that RGCs are the most vulnerable cells in the retina and the first cells to be damaged in DR [79,80]. Moreover, high levels of glucose have been correlated with thinning of the ganglion cell layer and expression of neuroinflammatory markers have been found in DR, leading us to consider damaged RGCs a crucial link between DR and glaucoma [27,43,81].

The vulnerability of RCGs is linked to their intense metabolism supported by a lot of mitochondria [82]. On the one hand, oxidative stress caused by several factors, such as high glucose (HG), may lead to mitochondrial damage and affect the viability of the RGCs [83]. On the other hand, apoptotic pathways activated by mitochondrial damage is an important mechanism of oxidative stress [84]. In this context, RGCs try to maintain mitochondrial integrity by eliminating the damaged mitochondria through mitophagy [85], that, in turns, may impair cellular homeostasis by altering the balance between mitochondrial biogenesis and mitophagy [86]. Therefore, the regulation of mitophagy, which has been shown to have a role in ocular diseases, is considered of particular importance in preserving RGCs survival [83,87,88,89].

The first evidence of GLP-1R expression in RGCs was reported by Fu et al. in 2012 [90]. Since then, several studies have demonstrated the efficacy of activation of GLP-1R in protecting RGCs cells (Table 2). Liraglutide, a GLP-1 analogue, and exenatide, a “long-lasting” GLP-1 receptor agonist, protected RGC-5 cells from death induced by high glucose and H_2_O_2_ [83,90,91]. Firstly, activation of GLP-1R markedly decreases H_2_O_2_- and high-glucose-induced reactive oxygen species (ROS) generation, thus improving cell survival [83,91]. Both these drugs exert their protection by improving mitochondrial function. In particular, exenatide counteracts detrimental effects of high glucose by reducing Bax expression and increasing the expression of Bcl-2 in RGC-5 cells [90]. Moreover, treatment with exendin-4 maintains the ratio between proapoptotic Bax and antiapoptotic Bcl-2/Bcl-xL in the retina of type II diabetic Goto-Kakizaki (GK) rats [65]. Like exenatide, liraglutide maintains mitochondrial membrane potential when RGCs are exposed to H_2_O_2_ [91]. Liraglutide also promotes mitochondria generation, and reduces mitophagy, by decreasing the expression of PINK1 and Parkin, two mitophagy-related proteins which are upregulated by high glucose [83]. Liraglutide prevented the autophagic process induced by H_2_O_2_ and high glucose by decreasing the expression of LC3A/B, structural proteins of autophagosomal membranes [83,91,92]. Furthermore, under H_2_O_2_ treatment, liraglutide counteracts the decreased expression of GAP43, an axonal marker protein, thus preserving RGCs function [91].

Activation of microglia and astrocytes plays an important role in the pathogenesis of glaucoma [73]. Interestingly, the DPP-IV inhibitor linagliptin and the long-acting GLP-1R agonist NLY01 were able to reduce activation of microglia during gliosis in STZ-diabetic rats [60,62]. In particular, NLY01 reduces the release from the microglia of mouse of proinflammatory cytokines (C1q, TNF-α, and IL-1α) involved in the progression of glaucoma [53,75,76,77,93]. Moreover, it has also been shown that expression of genes associated with neurotoxic astrocytes is significantly downregulated in C57BL6 mice treated with NLY01 [94]. Finally, NLY01 reduces the loss of RGC in a GLP-1R-dependent manner in mice [53].

Loss of RGCs may be also prevented by inhibition of Müller cell gliosis, a reactive phenotype characterized by increased expression of glial fibrillary acidic protein (GFAP) and production of proinflammatory factors [95,96]. In this context, treatment with exendin-4 drastically reduced the expression of GFAP in GK rats, thus reducing retinal gliosis [65]. The same goal has been achieved by treatment with geniposide, another agonist of GLP-1R, both in primary cultures of Müller cells and in the retina of mice with DR [97]. Moreover, treatment of Müller cells with exendin-4 decreased the expression of both placental growth factor and VEGF-A induced by high glucose exposure [47].

## 5. RPE Cells

Retinal pigment epithelium (RPE) is a monolayer of highly specialized hexagonal cells that forms the BRB [98,99]. The apical side of the RPE is connected to the outer segments of the photoreceptor (POS) through microvillous structures [98,99]. The lateral side of the cells are rich in tight junctional complexes which form a physical barrier that controls the diffusion of substances from the choroid into the subretinal space [98,99]. The basal side of the RPE is connected to the Bruch’s membrane, which separates RPE cells from the choroid. RPE plays important functions for the homeostasis of the retina and maintenance of normal vision, such as light absorption and protection against photo-oxidation, POS phagocytosis, ion and fluid transport, retinal barrier, secretion of cytokine and growth factors [100]. Therefore, alterations in the RPE are involved in the pathogenesis of several ocular disease including DR, AMD, and RP [98,99].

It is well known that hyperglycemia induces inflammatory responses in the retina, leading to several changes that include increased expression of adhesion molecules (ICAM-1 and VCAM-1), with consequent increment of monocyte adhesion, upregulation of VEGF-A secretion, metalloproteinases release and rise in ROS production [101,102,103]. Several studies demonstrated the protective effects of GLP-1R activation in RPE (Table 2), in particular that exenatide is able to counteract changes caused by inflammatory factors. Indeed, exenatide improves viability of RPE cells exposed to TNF-α and high glucose [102,103,104]. Moreover, in RPE cells treated with TNF-α, it reduces expression of VCAM and ICAM-1 [101], and levels of metalloproteinases (MMPs) and, contextually, increases the expression of its inhibitor, TIMP-2 [102]. Treatment with Exendin-4 has been also able to reduce H_2_O_2_-induced ROS production and to potentiate the antioxidant response in ARPE-19 cells [105]. This was achieved by increasing expression of nuclear factor erythroid 2-related factor-2 (NRF2), which, regulating expression of heme oxygenase-1 and NAD(P)H:quinone oxidoreductase-1, plays an important role in counteracting oxidative stress in several cell types [105,106]. Finally, geniposide inhibits hypoxia-induced activation of NF-kB, a key regulator of the inflammatory response, in ARPE-19 cells [107].

Hyperglycemia leads also to decreased expression of GLP-1R in streptozotocin-treated mice and in the human RPE cell line ARPE-19 [103]. Reduction in GLP-1R expression in ARPE-19 cells leads to increased intracellular ROS production and p53 expression [103], thus affecting RPE cell function and worsening the deleterial effects of hyperglycemia. In contrast, no differences in GLP-1R expression have been found between RPE from diabetic or healthy donors [48]. On the other hand, the bioavailability of GLP-1 may be reduced in diabetic subjects, indeed the levels of DPP-IV, the enzyme responsible of GLP-1 inactivation, have been found increased in RPE from diabetic donors [67] and in ARPE-19 cells exposed to TNF-α [102]. Treatment with exenatide was able to reduce the activity of DPP-IV induced by TNF-α [102].

## 6. Endothelial Cells and Pericytes

The retina has two vascular systems: the central retinal artery, that splits up into three capillary layers, and the choroid, which lies under the retinal pigment epithelium [108].

Microvascular retinal damage is mainly attributed to oxidative stress. The maintenance of the redox homeostasis is based on the balance between ROS production and antioxidant defenses. However, retinal endothelial cells are particularly susceptible to oxidative stress due to low amounts of the superoxide dismutase (SOD) [109]. It is well known that hyperglicemia induces overproduction of ROS and decreases the activity of antioxidative enzymes; consequently, oxidative stress is considered one of the main pathogenic factors in ocular complication of diabetes.

The positive vicious cycle that links inflammation to ROS production and cell apoptosis contributes to the damage of retinal vessels. Secretion of proinflammatory cytokines, such as IL-1β and TNF-α, is significantly increased in retinal endothelial cells exposed to HG. At the same time, HG decreases expression of GLP-1R [110]. Several studies reported that treatment with GLP-1RAs counteracts overproduction of ROS caused by high glucose (Table 2). In human retinal endothelial cells (HRECs), high glucose increases expression of sphingosine 1-phosphate (S1P) leading to activation of the S1P/sphingosine-1-phosphate receptor 2 (S1P/S1PR2) signaling, and consequent induction of ROS production and inflammation [110,111]. Treatment with exenatide decreases expression of S1PR2, thus inhibiting activation by S1P and decreasing ROS production [110]. Nian et al. reported that dulaglutide, a GLP-1 receptor agonist, counteracts the rise in ROS production induced by high glucose in HRECs [112]. At the same time, activation of GLP-1R may also increase the antioxidant defense. Indeed, exenatide induces the expression of SOD in HRECs [113], while dulaglutide increases the antioxidant defenses by restoring the activity of Gluthatione peroxidase [111].

Oxidative stress may also accelerate endothelial cell senescence, thus affecting their viability, by reducing expression of Sirtuin-1 (SIRT-1), which has antioxidative and antiaging activity [114]. Treatment with dulaglutide restores the levels of SIRT-1 and the activity of telomerase, thus reducing cell senescence induced by high glucose [112]. Moreover, oxidative stress due to high glucose may induce apoptosis of HRECs [110]. Treatment with exenatide counteracts HG-induced cell apoptosis, increasing expression of Bcl2 and reducing expression of Bax and activation of caspases [110].

Retinal endothelial cells contribute to the formation of the BRB. Therefore, their damage may compromise the BRB function, leading to increased vascular leakage, which is a pathogenic factor in several ocular diseases such as such as DR, AMD, retinal vein occlusion and uveitis. In this context, the inhibitory effects of exenatide on the S1P/S1PR2 pathway is of particular importance, since this signaling pathway affects vascular barrier function [111]. Changes in permeability of the BRB are attributed to increased secretion of the VEGF-A, which is considered the main inducer of ICAM-1 in early phase of DR [115]. It has been reported that activation of GLP-1R may interfere with VEGF-A-mediated signaling in endothelial cells [116,117]. In particular, stimulation of HRECs with GLP-1 inhibits phosphorylation of Phospholipase Cγ (PLCγ) and Endothelial Nitric Oxide Synthase (eNOS) induced by VEGF-A, thus inhibiting VEGF-A-induced vasodilatation [116]. On the other hand, the maintenance of the retinal capillary tone is important in ischaemia/reperfusion (I/R) injury, which can occur in several ocular diseases. In this event, exendin-4 may contribute to reversing contraction of capillaries by increasing eNOS phosphorylation [69]. Moreover, dulaglutide prevents downregulation of eNOS expression in HRECs exposed to HG [112].

It has been shown that vessel permeability is directly correlated with the density of pericytes, and loss of pericytes is considered an early feature of DR [118]. In the retinal vessels, the ratio between pericytes and endothelial cells is higher as compared to other microvascular compartments. Expression of GLP-1R on pericytes has been poorly investigated, and there are contradictory results [69,119]. However, it has been found that, in vitro, liraglutide attenuates the migration of retinal pericytes induced by advanced glycation end products [119], suggesting that GLP-1R may be activated also in in pericytes, and may prevent their loss in DR.

## 7. Conclusions

Inflammation is considered one of the main causes of retinal dysfunction. In this review, we explored the effects of GLP-1R activation in the retina, showing that treatment with GLP-1, GLP-1RAs or DPP-IV inhibitors leads to prevention or improvement of retinal inflammatory diseases (Figure 1).

Oxidative stress is a common trigger factor in pathogenesis of different inflammatory retinal diseases, especially in hyperglycemic condition. Increased production of ROS, in turn, may contribute to further worsening inflammatory states by increasing release of proinflammatory cytokines. The onset of this vicious cycle may lead to cell dysfunction and, finally, to cell death. Activation of GLP-1R may counteract oxidative stress by decreasing ROS production [70,83,91,103,105,110,112], increasing antioxidant defense [70,106,112,113] and improving mitochondrial function maintaining both mitogenesis [83,91] and membrane permeability [65,90,110].

The breakdown of BRB may further contribute to increasing the inflammatory state of the retina. Therefore, preserving its integrity should be a common goal of all therapies. Several evidence demonstrated that activation of GLP-1R may improve BRB function, thus maintaining vessel permeability, by acting at different levels: decreasing the migration of retinal pericytes [62,119]; decreasing secretion of VEGF-A and expression of ICAM and VCAM [101], preventing downregulation of tight junctions [65].

However, most of this evidence comes from studies performed in vitro or in animal models, thus limiting the evidence that GLP-1R activation may have beneficial effects also in human retinal inflammatory diseases. Indeed, although clinical trials investigated the safety of GLP-1R activation in humans, in particular in diabetic subjects, their major interest has been focused on cardiovascular outcomes, without considering the impact on DR [54,55,56,57,58,120].

## 8. Future Perspectives

Evidence that activation of GLP-1R may have beneficial effects in retinal inflammatory diseases, also without affecting glycemic control, suggests that it may be useful in ocular pathologies not necessarily related to diabetes. In this scenario, topical administration of GLP-1RAs or DPP-IV inhibitors could represent an effective strategy to obtain specifically retinal beneficial effects.

## Figures and Tables

**Figure 1 ijms-23-12428-f001:**
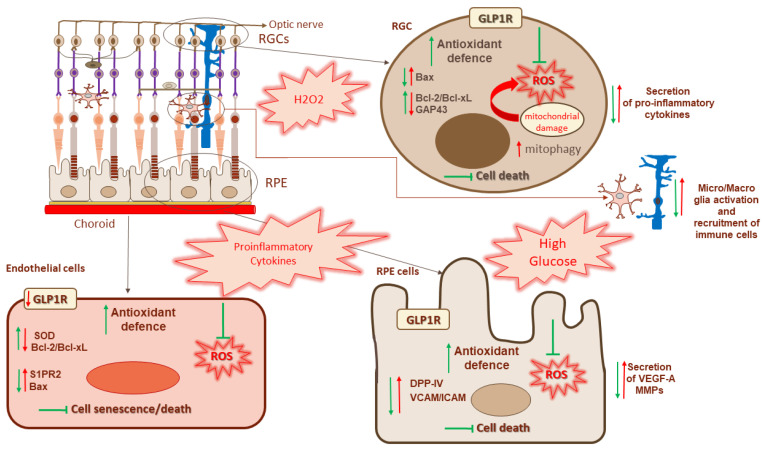
Schematic representation of retinal layers with highlight on retinal ganglion cells (RGCs), retinal pigment epithelium (RPE) and choroid. Overview of the main detrimental effects of proinflammatory factors (cytokines, high glucose concentration and H_2_O_2_) are represented in red with arrows and text in each cell type. The effects of GLP-1R activation are represented in green.

**Table 1 ijms-23-12428-t001:** Brief description of GLP-1RAs cited in the text.

Drug	Class	Description
exendin-4	GLP-1RA	39 amino acid peptide found in the saliva of the Gila monster (Heloderma suspectum), with 53% homology to human GLP-1 and resistance to DPP-4 inactivation.
exenatide	GLP-1RA	Synthetic analog of exendin-4 resistant to the proteolytic effect of DPP-4
liraglutide	GLP-1RA	Modified human GLP-1(7–37) with a 97% homology and longer half-life (13 h).
semaglutide	GLP-1RA	Long-acting GLP-1 agonist with a 94% homology and a half-life of 1 week.
dulaglutide	GLP-1RA	Two GLP-1 molecules linked to an IgG4-Fc heavy chain and with a half-life of 4 days.
NLY01	GLP-1RA	Pegylated exendin-4 analogue with an extended half-life.
geniposide	GLP-1RA	Iridoid glycoside, a natural compound found in several medicinal herbs.
linagliptin	DPP-IV inhibitor	

**Table 2 ijms-23-12428-t002:** Effects of GLP-1R activation in the retinal cells injured by inflammation, oxidative stress and high glucose.

	Cell Type	RGCs	RPE Cells	Endothelial Cells
Drugs				
exendin-4		Mantains ratio between Bax and Bcl2/Bcl-xL	↓ ROS production Increases the antioxidant response	↑ eNOS phosphorylation
exenatide		Improves mitochondrial function ↓ Bax ↑ Bcl-2	Improves viability ↓ VCAM, ICAM-1 and MMPs ↑ TIMP-2 ↓ DPPIV activity	↓ ROS production, caspase activation and apoptosis Induces expression of SOD ↓ Bax ↑ Bcl-2
liraglutide		Improves mitochondrial function ↓ mitophagy and autophagy Promotes mitochondrial generation Maintains expression of the axonal marker GAP43		
dulaglutide				↓ ROS production Restores Gluthatione activity, levels of SIRT-1 and telomerase activity Prevents reduction in eNOS expression
NLY01		↓ release of proinflammatory cytokines by microglia ↓ expression of genes associated with neurotoxic astrocytes		
geniposide				↓ NF-kB activation

## Data Availability

Not applicable.

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
