# Peer review of "Anti-Inflammatory Effects of GLP-1R Activation in the Retina"

_ijms, 2022, doi:10.3390/ijms232012428_

Round 1

Reviewer 1 Report

The article titled: “ Anti-inflammatory effects of GLP-1R activation in the retina”  by A. Puddu et al.  summarizes   the anti-inflammatory effects of GLP-1R activation in the retina.

The review shows  the pathogenic role of inflammation in ocular diseases and it describes the pleiotropic effects of GLP-1R activation on the cellular components of the retina which are mainly involved in the pathogenesis of inflammatory retinal diseases.

The authors have described how GLP-1 and/or its receptor participate in one way or another in the development of retinal degeneration. However, during the development of the article, they have been alternately including paragraphs dedicated to glaucoma, RP, uveitis, among others, with DR being the main subject of analysis. Indeed, the number of studies dedicated to one type of pathology or another is disparate. If the authors do not consider expanding the information about these other pathologies not associated with Diabetes, they should consider not including the scarce information.

Paragraph 5 needs to be restructured. The signaling pathways modulated by two treatments are discussed. It is recommended to first expose the affected pathways, and then comment on the key proteins that modulate the treatments. A possible comparative table would give a clearer view of the effects of each of the treatments.

The expression of GLP1R is fundamentally in the RGC, however recent studies show that part of the retinal immune system has the ability to respond to GLP1 analogues.

In Figure 1, the presence of macro and microglia within the anti-inflammatory or pro-inflammatory response that occurs in DR is missing.

Reference 90 comments on the response of the RCG-5 cells, and it should be noted that since 2014 it is not allowed to use this cell type since its origin has been much discussed and it seems that they do not have the characteristics of the cell type at which represent.

Author Response

The article titled: “Anti-inflammatory effects of GLP-1R activation in the retina” by A. Puddu et al.  summarizes   the anti-inflammatory effects of GLP-1R activation in the retina.

The review shows the pathogenic role of inflammation in ocular diseases, and it describes the pleiotropic effects of GLP-1R activation on the cellular components of the retina which are mainly involved in the pathogenesis of inflammatory retinal diseases.

Replay: We are grateful to Reviewer #1 for his/her effort reviewing our paper and his/her positive feedback. The summary of our work as written by this reviewer is precise. Here we address the questions and suggestions raised by the reviewer #1.

The authors have described how GLP-1 and/or its receptor participate in one way or another in the development of retinal degeneration. However, during the development of the article, they have been alternately including paragraphs dedicated to glaucoma, RP, uveitis, among others, with DR being the main subject of analysis. Indeed, the number of studies dedicated to one type of pathology or another is disparate. If the authors do not consider expanding the information about these other pathologies not associated with Diabetes, they should consider not including the scarce information.

Replay: The Reviewer's observation is correct. The discussion of diabetic retinopathy is more in-depth because the topic of the review is the activation of GLP-1R, which is mainly studied in diabetic models. Therefore, most of the information available in the literature is on diabetic retinopathy rather than on other inflammatory diseases of the eye. However, our aim was to summarize the pleiotropic effects of GLP-1R activation in inflammatory retinal diseases. In our opinion, it is important that this review is not limited to discuss only diabetic retinopathy, but also provides the state of the art in other diseases, precisely because information on these is scarce.

Paragraph 5 needs to be restructured. The signaling pathways modulated by two treatments are discussed. It is recommended to first expose the affected pathways, and then comment on the key proteins that modulate the treatments. A possible comparative table would give a clearer view of the effects of each of the treatments.

Replay: We thank you the reviewer for his/her advice. We have revised paragraph 5 in according to the reviewer's advice.

Retinal pigment epithelium (RPE) is a monolayer of highly specialized hexagonal cells that forms the BRB [98, 99]. The apical side of the RPE is connected to the outer segments of photoreceptor (POS) through microvillous structures [98, 99]. The lateral side of the cells are rich of tight junctional complexes which form a physical barrier that controls the diffusion of substances from the choroid to the subretinal space [98, 99]. The basal side of the RPE is connected to the Bruch’s membrane which separates RPE cells from the choroid. RPE plays important functions for the homeostasis of the retina and maintenance of normal vision, such as light absorption and protection against photo-oxidation, POS phagocytosis, ion and fluid transport, retinal barrier, secretion of cytokine and growth factors [100]. Therefore, alterations in the RPE are involved in the pathogenesis of several ocular disease including DR, AMD, and RP [98, 99].

It is well known that hyperglycemia induces inflammatory response in the retina, leading to several changes that include increased expression of adhesion molecules (ICAM-1 and VCAM-1), with consequent increment of monocyte adhesion, up-regulation of VEGF-A secretion, metalloproteinases release and rise in ROS production [101-103]. Several studies demonstrated the protective effects of GLP-1R activation in RPE, in particular that exenatide is able to counteract changes caused by inflammatory factors. Indeed, exenatide improves viability of RPE cells exposed to TNF-α and high glucose [102-104]. Moreover, it reduces expression of VCAM and ICAM-1 [101] and levels of metalloproteinases (MMPs) and, contextually, increases the expression of its inhibitor, TIMP-2, in RPE cells treated with TNF-α [102]. Treatment with Exedin-4 has been also able to reduce H2O2-induced ROS production and to potentiate the antioxidant response in ARPE-19 cells [105]. Finally, Geniposide inhibits hypoxia-induced activation of NF-kB, a key regulator of the inflammatory response, in ARPE-19 cells [106].

Hyperglycemia leads also to decreased expression of GLP-1R in Streptozotocin-treated mice and in the human RPE cell line ARPE-19 [103]. Reduction of GLP-1R expression in ARPE-19 cells leads to increased intracellular ROS production and p53 expression [103], thus affecting RPE cell function and worsening the deleterial effects of hyperglycemia. In contrast, no differences in GLP-1R expression has been found between RPE from diabetic or healthy donors [48]. On the other hand, the bioavailability of GLP-1 may be reduced in diabetic subjects, indeed the levels of DPP-IV, the enzyme responsible of GLP-1 inactivation, have been found increased in RPE from diabetic donors [67] and in ARPE-19 cells exposed to TNF-α [102]. Treatment with exenatide was able to reduce the activity of DPP-IV induced by TNF-α [102].

Furthermore, we added a comparative table (Table 2) to make clearer the anti-inflammatory effects of the drugs in each cell type.

The expression of GLP1R is fundamentally in the RGC, however recent studies show that part of the retinal immune system has the ability to respond to GLP1 analogues.

Replay: The ability to modulate immune system in the retina is an important mechanism through which activation of GLP-1R regulates the inflammatory response in retinal diseases. According with your observation, we discussed this topic at the end of the Section 4: “Activation of microglia and astrocytes plays an important role in the pathogenesis of glaucoma [73]. Interestingly, the DPP-IV inhibitor Linagliptin and the long-acting GLP-1R agonist NLY01 were able to reduce activation of microglia during gliosis in STZ-diabetic rats [60, 62]. In particular, NLY01 reduces the release from the microglial of mouse of pro-inflammatory cytokines (C1q, TNF-α, and IL-1α) involved in the progression of Glaucoma [53, 75-77, 93]. Moreover, it has also been shown that expression of genes associated with neurotoxic astrocytes, are significantly downregulated in C57BL6 mice treated with NLY01 [94]. Finally, NLY01 reduces the loss of RGC in a GLP-1R-dependent manner in mice [53]

In Figure 1, the presence of macro and microglia within the anti-inflammatory or pro-inflammatory response that occurs in DR is missing.

Replay: We added the effects of GLP-1R activation on macro- and microglia to Figure 1

Reference 90 comments on the response of the RCG-5 cells, and it should be noted that since 2014 it is not allowed to use this cell type since its origin has been much discussed and it seems that they do not have the characteristics of the cell type at which represent.

Replay: We thank the Reviewer for this observation. The origin of RGC-5 cells has been object of great discussion in the literature. In 2014 C. Sippl and E. R. Tamm summarized the question regarding the origin and nature of the RGC-5 cells (C Sippl, E R Tamm What is the nature of the RGC-5 cell line? Adv Exp Med Biol. 2014; 801:145-54. doi: 10.1007/978-1-4614-3209-8_19). However, in 2017 Sayyad et all investigated the expression of markers specific to retinal ganglion cells in RGC-5 cell line demonstrating that this cell line showed the expression of essentially the same markers of retinal ganglion cells (Sayyad  et al. 661W is a retinal ganglion precursor-like cell line in which glaucoma-associated optineurin mutants induce cell death selectively. Sci Rep. 2017 Dec 4;7(1):16855.doi: 10.1038/s41598-017-17241-0). Sayyad et all concluded that it is reasonable to suggest that these cells may be useful as a cell culture model for investigating the molecular mechanisms involved in induction of cell death relevant for glaucoma pathogenesis. Reference 90 is an article published in 2012, and it has not been retracted. Probably the conclusions of Sayyad justify the use of RGC-5 cells in the last decade. Indeed, in this review we cited articles published after 2014 in which RGC-5 cells have been used as a model of retinal ganglion cells. None of them has been retracted so we have cited them.

Reviewer 2 Report

In this review, the authors have put in great effort in describing the anti-inflammatory effects of GLP-1R activation in the retina. Although I do feel that if it would be beneficial to the readers, if the authors could provide with some tables describing the different drugs mentioned in the review. Some of other suggestions are mentioned below:

Line 65.. It would be better if the authors specify or give examples of other diseases wherein GLP-1 signaling plays a role. 

Line 76..needs more than just one ref for that sentence. 

Lines 115-116... The authors state that several experimental research demonstrated that GLP-1 and its agonist may prevent onset of DR. It would be good if they specify what kind of research...in vitro or in vivo...the sentence needs more clarification. Stating actual studies will be more relevant to the review instead of just one sentence. 

Line 161...Damaged RGCs are considered crucial link between DR and glaucoma... could the authors be more specific as to in what context?

It would be nice if the authors can tabulate information of all the drugs used for specific cell types in the retina  mentioned in the review, specifically describing the anti-inflammatory effects of GLP-1 in the retina. 

Line 192-197.... Describe what kind of mice were used in the study 

Line 203-205.. Give more info on the drug exedin-4, geniposide..how are they related to GLP-1R.

Line 333..The authors mention that topical administration of GLP-1RAs or DDP-IV inhibitors ... are there studies have shown they can be used topically??

Author Response

In this review, the authors have put in great effort in describing the anti-inflammatory effects of GLP-1R activation in the retina. Although I do feel that if it would be beneficial to the readers, if the authors could provide with some tables describing the different drugs mentioned in the review.

Replay: We are grateful to Reviewer #2 for his/her effort reviewing our paper and his/her positive feedback. Here below we address the questions and suggestions raised by the Reviewer #2.

Some of other suggestions are mentioned below:

Line 65.. It would be better if the authors specify or give examples of other diseases wherein GLP-1 signaling plays a role.

Replay: According to your suggestion we specified some of other diseases: “These evidence support that GLP-1 signaling plays an important role not only in the management of diabetes, but also in the treatment of other pathologies, including neuronal and cardiovascular diseases, such as Parkinson’s Disease and atherosclerosis [24]….”

Line 76..needs more than just one ref for that sentence.

Replay: We added other references to this sentence.

Lines 115-116... The authors state that several experimental research demonstrated that GLP-1 and its agonist may prevent onset of DR. It would be good if they specify what kind of research...in vitro or in vivo...the sentence needs more clarification. Stating actual studies will be more relevant to the review instead of just one sentence.

Replay: According to your suggestion we specified the sentence: “…. Several experimental research demonstrated that GLP-1 and its agonists may prevent the onset of DR or reduce its progression through antiapoptotic and anti-inflammatory mechanisms [47-51]. Exedin-4 prevents BRB breakdown, and decreases levels of ICAM-1 in retina of Goto-Kakizaki(GK) rats and decreases the expressions of PLGF and VEGF and the activation of AKT in vitro [47]. Moreover, treatment with Exedin-4 decreased retinal cell death and ROS production in Streptozotocin rats and protects retinal precursor cells from oxidative stress [51]. Both systemic and topical administration of native GLP-1 or GLP-1RAs prevents glial activation and apoptosis in retina of db/db mice [48].…. ”

Line 161...Damaged RGCs are considered crucial link between DR and glaucoma... could the authors be more specific as to in what context?

Replay: It is well known that people with diabetes are twice as likely to develop the so call open-angle glaucoma (the most common type of glaucoma) as are non-diabetics. On the other hand, the abnormal blood vessels growth in DR can lead to increased intraocular pressure and the onset of the so called neovascular glaucoma (34495413). Diabetes and glaucoma share common pathophysiological mechanisms, such as glial cell activation, inflammation, and neuronal degeneration, which may lead respectively to neuropathy and RGC depletion. High levels of glucose have been correlated with thinning of the ganglion cell layer in the retina. Furthermore, expression of neuroinflammatory markers have been found in DR. Therefore, progressive damage of RGCs during DR may increase the occurrence of glaucoma. In light of these consideration, we re-wrote the sentence: “Due to their importance in visual production, damage of the RGCs may lead to visual impairment and even to permanent loss of vision [78]. In diabetes, neurodegeneration occurs earlier than the microvascular injury of the retina, and it is well recognized that RGCs are the most vulnerable cells in the retina and the first cells to be damaged in DR [79, 80]. Moreover, high levels of glucose have been correlated with thinning of the ganglion cell layer and expression of neuroinflammatory markers have been found in DR, leading to consider damaged RGCs a crucial link between DR and GLAUCOMA [42, 43, 81].”

It would be nice if the authors can tabulate information of all the drugs used for specific cell types in the retina  mentioned in the review, specifically describing the anti-inflammatory effects of GLP-1 in the retina.

Replay: We added a comparative table (Table 2) to make clearer the anti-inflammatory effects of the drugs in each cell type.

Line 192-197.... Describe what kind of mice were used in the study

Replay: The C57BL6 mice have been used in this study, we inserted this information in the text: “….Moreover, it has also been shown that expression of genes associated with neurotoxic astrocytes are significantly downregulated in C57BL6 mice treated with NLY01 [94]…”

Line 203-205.. Give more info on the drug exedin-4, geniposide..how are they related to GLP-1R.

Replay: we added a table (Table 1) with the main features of the drugs cited in this review.

Line 333..The authors mention that topical administration of GLP-1RAs or DDP-IV inhibitors ... are there studies have shown they can be used topically??

Replay: To our knowledge topical administration of GLP-1Ras or DPPIV inhibitors has been performed only in animal models (Hernandez C. et al. Topical Administration of GLP-1 Receptor Agonists Prevents Retinal Neurodegeneration in Experimental Diabetes. Diabetes 2016, 65, (1), 172-87; Hernandez C. et al. Topical administration of DPP-IV inhibitors prevents retinal neurodegeneration in experimental diabetes. Diabetologia 2017, 60, (11), 2285-2298; Xingsheng Shu, et al. Topical ocular administration of the GLP-1 receptor agonist liraglutide arrests hyperphosphorylated tau-triggered diabetic retinal neurodegeneration via activation of GLP-1R/Akt/GSK3β signaling. Neuropharmacology 2019 Jul 15;153:1-12. doi:10.1016/j.neuropharm.2019.04.018). In these models topical administration of GLP-1RAs or DDP-IV inhibitors was effective in preventing neurodegeneration and vascular leakage in the diabetic retina. These successful results may encourage the employment of these drugs in the treatment of early diabetic retinopathy.

Round 2

Reviewer 1 Report

The authors have considered all previous suggestions. The article is ready to be accepted.

Author Response

We thank you for your positive feedback

Reviewer 2 Report

I believe the authors have addressed previous concerns and improved the work. 

Author Response

We thank you for your positive feedback